# Multi-Response Robust Parameter Optimization of Cemented Backfill Proportion with Ultra-Fine Tailings

**DOI:** 10.3390/ma15196902

**Published:** 2022-10-05

**Authors:** Mingqing Huang, Sijie Cai, Lin Chen, Shaohui Tang

**Affiliations:** 1Zijin School of Geology and Mining, Fuzhou University, Fuzhou 350108, China; 2State Key Laboratory of Comprehensive Utilization of Low-Grade Refractory Gold Ores, Longyan 356214, China

**Keywords:** orthogonal design, response surface method, robust parameter, ultra-fine tailings cemented filling, optimization proportion

## Abstract

Backfill of mined-out areas in Carlin-type gold mines always encounters the challenges of ultra-fine tailings, low backfill strength and difficult slurry transportation caused by fine tailings. To understand the influence of slurry mass concentration, waste rock content, and cement-sand ratio on the cemented backfill strength and fluidity, influential factors were determined by range analysis of orthogonal proportion experiments. Response surface methodology (RSM) was used to analyze the influence of each factor on response, and the backfill strength and slump were optimized using a robust optimization desirability function method. The results show that the cement-sand ratio has the highest effect on the backfill strength, and the slurry slump is dominated by the slurry mass concentration. The interaction between waste rock content and the cement-sand ratio significantly impacts the slump, while the interaction between the slurry mass concentration and the cement-sand ratio has a positive correlation with the backfill strength. The ultra-fine tailings cemented backfill proportion was optimized by using multi-response robust parameters as 68.36% slurry mass concentration, 36.72% waste rock content and 1:3 cement-sand ratio. The overall robust optimal desirability was 0.8165, and the validity of multi-response robust parameter optimization was verified by laboratory tests.

## 1. Introduction

With the continuous depletion of open-pit and shallow metal mineral resources, many mines have turned to deep mining. Cut and fill mining refers to a mining method that first cuts the stope room and then fills the mined-out areas with filling materials. Cut and fill mining plays an irreplaceable role in deep and green mining since it utilizes a bulk of solid wastes such as tailings and waste rocks. A successful application of backfill mining means an efficient substitute of ores with backfill materials with a certain strength and low cost. However, filling cost, unconfined compressive strength of backfill (backfill strength) and fluidity are closely related to the proportion of filling materials and mass concentration of slurry [1,2].

Backfill strength is an important parameter for mining environmental stability and a key indicator for assessing the applicability of safe and continuous mining [3]. A low backfill strength tends to fail in sufficient support for stopes and underground engineering, while an excess backfill strength leads to high cement consumption and operation costs. Therefore, control of backfill strength, filling cost and their optimization by adjusting filling proportion to meet the mining requirements become a consensus [4]. Filling proportion not only affects the backfill strength, but also plays an important role in slurry flow characteristics, filling cost and filling process decision [5].

Wang et al. [6] studied the sensitivity of tailings particle size, cement-sand ratio and slurry mass concentration on backfill strength through range analysis, variance analysis and regression analysis. The regression model between backfill compressive strength and the influential factors predicted the variation law of backfill compressive strength. Yu et al. [7] studied the influence of waste rock content, concentration and cement-sand ratio on the cemented backfill strength by RSM-BBD, and application has been achieved in field practice. Gao et al. [8] studied the influential factors of mixed aggregate cemented backfill through range analysis and variance analysis and established a regression model that can effectively predict the change law of slump and compressive strength of mixed aggregate cemented backfill. Likewise, Qi et al. [9] analyzed the sensitivity influential factors of early backfill strength with Expert Design 12.0 software and proposed an optimal proportion of backfill cementing materials through variance analysis and regression optimization. Deng et al. [10] used tailings, ordinary Portland cement and additives as filling materials to study the strength law of cement tailings backfill under different curing conditions. Zhu et al. [11] studied the influence of different ratios on the rheological properties of cemented coal gangue backfill using the RSM method and established a quadratic polynomial regression model of yield stress and plastic viscosity. In addition, Cao et al. [12] conducted an experimental study on the particle size distribution and chemical composition of gold mine tailings and studied the relationship between tailings cementing slurry concentration, waste rock content and cement-sand ratio through viscosity and uniaxial compressive strength tests. Petlovanyi et al. [13] studied the expediency of applying binder material mechanical activation in a cemented rockfill (CRF). The polynomial dependences have been obtained of the strength variation of the CRF. Mashifana et al. [14] used the green process of non-radioactive gold tailings and granular blast furnace (GBF) slag to synthesize a green and sustainable filling material. The results show that the particle size distribution (PSD) had a significant impact on the strength of GBFS-modified GT. Petlovanyi et al. [15] studied the strength and microstructure properties of backfill. The results showed no strong low-basic hydrated calcium silicate bonds that could have a reinforcing effect. Recently, Huang et al. [16] investigated the influence of law on the strength of neutralized slag-cemented backfill and determined the optimal proportion combination from the perspective of economy and engineering by using RSM-BBD design and multi-objective function optimization; however, the robustness that affects the performance of the backfill is not considered. Many scholars have studied the optimization of filling ratios and proposed informative results. However, previous research on filling ratio optimization did not consider the factor of robustness. In the filling ratio, the stability or sensitivity of the working performance of the backfill is important when the content of the backfill components (such as cement, ultra-fine tailings, waste rock) fluctuates [17,18,19]. This is mainly caused by the inaccuracy of the filling material composition and the changes in the characteristics in the production process. Therefore, in this paper, an orthogonal test and RSM were used to study and analyze the influential factors of backfill strength and slump, as well as the interaction between influential factors. On this basis, a robust desirability function method was proposed to determine the optimal matching scheme, weighing the optimality and robustness.

Carlin-type gold deposits are widespread worldwide and refractory to mineral processing. These deposits are dominated by carbonate rock sedimentary rock as host rock; gold minerals exist within ore particles and are ultra-fine. Thus, ores are characterized by fine disseminated minerals, which need to be ground into ultra-fine particles during beneficiation to separate from the host rock by pressurized oxidation. This grinding behavior, however, will produce a large number of ultra-fine tailings. As a common backfill material, ultra-fine tailing always shows a poor cementitious effect with binding materials. Shuiyindong gold mine, a large Carlin-type gold deposit located in southwest China, utilizes resources by the cut and fill mining method. For years, its ultra-fine tailings and poor granular composition have led to problems of low backfill strength and difficult slurry transportation. The objective of this work is to investigate the influence of slurry mass concentration, waste rock content, cement-sand ratio and their interactive influence on cemented backfill strength and fluidity with ultra-fine tailings using orthogonal proportion experiments and response surface methodology. First, an orthogonal experimental design was used to study the relationship between single factor and ultra-fine cemented backfill strength and slurry flow characteristics and determine the primary sequence of influential factors. Second, RSM was introduced to study the relationship between the interaction of multiple factors and the ultra-fine cemented backfill strength and slurry flow characteristics, and the validity of the conclusion of the orthogonal test was verified. Finally, the optimal ratio of backfill strength and slump was determined by the robust optimization desirability function method, and its reliability was verified by laboratory tests.

## 2. Materials and Methods

### 2.1. Materials

The materials included waste rock, ultra-fine tailings and M32.5 Portland cement. The waste rocks in the test were sampled from waste rock piles of the Shuiyindong gold mine. The main types of waste rock are clastic limestone, breccia and calcareous sandstone. The random sampling method was used for on-site sampling and measurement. The results of the waste rock particle size composition are shown in Table 1.

When the particle size of the waste rock < 5 mm, the transportation performance of the aggregate is better and reduces the wear of the pipeline. Thus, a particle size below −5 mm was selected for the test. In the laboratory, we used a vibration crusher or ball mill to crush the waste rock to the size of −5 mm. In the lab tests, the water content of waste rock was 0.12%, the density was 2.67 g·cm^−3^, the compaction density was 1.80 g·cm^−3^ and the loose density was 1.62 g·cm^−3^.

Ultra-fine tailings are derived from tailing mortar discharged from the concentration plant of Shuiyindong gold mine. After drying, the test results show that the average density of ultra-fine tailings is 2.684 g·cm^−3^, the loose density is 1.03 t·m^−3^ and the porosity is 61.6%, the compaction density is 1.37 t·m^−3^ and the porosity is 48.8%. NITON XL3T950 X-ray fluorescence spectrometry (XRF), chemical element calibration method, LS-CWM(3) laser particle size analyzer, and the focused beam reflection measurement (FBRM) method were used to analyze the basic chemical composition and particle size distribution of ultra-fine tailings. The results are shown in Table 2 and Figure 1.

According to the tailing particle size distribution curve, the characteristic parameters of particle size can be calculated as d_10_ =3.41 μm, d_30_ =6.65 μm, and d_60_ =13.16 μm, respectively. According to Equation (1), the non-uniformity coefficient C_u_ was 3.86 and the curvature coefficient C_c_ was 1.36. This indicates that the tailings have a narrow particle size distribution range and a poor continuous distribution. A large proportion of ultra-fine tailings usually make cemented backfill difficult to dehydrate and increase pipeline transportation resistance [20,21].
(1)$$\left\{\begin{array}{c} C_{u}{=d}_{60}{/d}_{10}\\ C_{c}{=d}_{30}{}^{2}{/(d}_{60}d_{10}) \end{array}\right.$$

The cementing material is M32.5 ordinary Portland cement (Zhenfeng rendu cement Co., Ltd., Qianxinan, China). When a standard cubic block (7.07 cm × 7.07 cm × 7.07 cm) is cured for 28 days at 20 ± 2 °C and relative humidity above 90%, the unconfined compressive strength of the M32.5 cement mortar block reaches 32.5 MPa. Its density was 3.02 g/cm^3^. The chemical composition is mainly Al_2_O_3_, SiO_2_ and CaO. The cement is characterized by high late strength, low hydration heat and good water retention.

### 2.2. Orthogonal Experiment of Filling Proportion

A three-factor and five-level orthogonal design scheme was introduced to test the influence of variable factors on backfill strength and fluidity. Slurry mass concentration, waste rock content and cement-sand ratio were taken as the influential factors. The 7-day and 28-day uniaxial compressive strength and slump of backfill were taken as the objective functions (Table 3). Range analysis was used to study the relationship between each factor and the objective function. There are 25 groups of orthogonal test schemes, as shown in Table 4.

### 2.3. RSM Experiment of Filling Proportion

The cemented backfill strength and slurry transportation are important factors in mine production and are closely related to the optimization of the filling proportion. Orthogonal experimental design is a popular fractional factorial design used in filling proportion optimization [22] and is beneficial for analyzing the influence of each factor on the target value [23]. Likewise, RSM is an optimization method that integrates experimental design and mathematical modeling, which can effectively reduce test runs and reveal the interactions among influential factors. Scholars adopted RSM to optimize the ratio of filling materials and studied the influential factors in different curing ages and the influential factors’ interaction on backfill strength [24]. Based on the desirability function, RSM can implement robust optimization of multi-objective [25] and analyze the optimal filling proportion test results. RSM shows advantages in uncovering relationships between the interaction of multiple factors and the objective function and can verify the analysis results of orthogonal tests. The robust parameter optimization method reduces the sensitivity of a process to noise factors by selecting the horizontal combination of controllable factors and makes the experimental target more stable and reliable.

A Box–Benhnken central composite experiment design was used to carry out the filling proportion test. It is mainly used to study the influence of multi-factor interaction on backfill strength and slump and can verify the reliability of orthogonal test results. The slurry mass concentration, waste rock content and cement-sand ratio were taken as variables, and the 7-day, 28-day uniaxial compressive strength and slump of the backfill were taken as response values. The influence of each variable of the filling proportion and its interaction on the strength and slump of the backfill was studied. Variance analysis and response surface analysis were used to analyze the influence of the interaction between variables of the mixed proportion in the response value. The design scheme is shown in Table 5. There are 17 groups of tests, and the scheme is shown in Table 6.

### 2.4. Experimental Process

According to the experimental design, the waste rock, ultra-fine total tailings, cement, water and water reducing agent (1~2.5% to cement weight) were added into the container and the uniform slurry was stirred by a mixer. The slump of the filling slurry was measured by a slump cylinder, whose bottom surface diameter was 200 mm, top surface diameter was 100 mm, and cylinder height was 300 mm. The process is shown in Figure 2. The slump test process of backfill materials is in accordance with the “Standard for Performance Test Methods of Ordinary Concrete Mixtures” (GB/T 50080-2002). Then, an evenly stirred slurry was loaded into the standard mold of 7.07 cm × 7.07 cm × 7.07 cm to allow natural settlement. After the specimen initially stood on its own for 24 h, demolding treatment was carried out. After demolding, the specimen was placed in the curing box for curing (the curing temperature was 20 °C, and the curing humidity was 90%). The QKX-ZSZ-4000 triaxial dynamic and static load test system (Qingdao qiankunxing intelligent Co., Ltd., Qingdao, China) was used to test the uniaxial compressive strength of backfill on days 7 and 28, and the average uniaxial compressive strength of the three specimens was taken as measured values. A total of 25 × 6 specimens were made. After curing for 7 days and 28 days, three specimens were taken from each group for testing and the average value of the three test results was taken as the calculation data. The process is shown in Figure 3. The process of backfill test specimens is according to the “Standard Test Method for Mechanical Properties of Ordinary Concrete” (GB/T 50081-2002).

## 3. Results and Discussion

### 3.1. Influence of Single Factors on Backfill Strength and Slump

The distribution diagram of orthogonal test results of filling proportion (Figure 4) shows that the compressive backfill strength at each curing age decreases with the decrease in the cement-sand ratio at the same slurry mass concentration. With the same cement-sand ratio, the backfill strength increases with the increase of slurry mass concentration. Meanwhile, the backfill strength increases with curing time. The addition of tailings and waste rock improves the filling concentration and backfill compressive strength. Waste rock has the highest influence on slurry slump. Adding waste rock into the ultra-fine tailings improves the slump and the fluidity under the condition of high concentration.

Range analysis was adopted for slurry mass concentration, waste rock content and cement-sand ratio to determine the main influential factors of backfill strength and slump at different curing ages (Table 7).

Table 7 shows that the maximum range between 7-day uniaxial compressive strength and 28-day uniaxial compressive strength is determined by the cement-sand ratio, which is 1.442 and 1.041, respectively, indicating that the cement-sand ratio has the highest influence on 7-day and 28-day backfill strength. The influential factors with the smallest range values are the waste rock content, which is 0.145 and 0.303, respectively, indicating that the waste rock content has the lowest influence on the 7-day and 28-day backfill strength. The influential factor sequence in the compressive backfill strength at 7 days and 28 days is the cement-sand ratio > slurry mass concentration > waste rock content. The cement-sand ratio is the main influential factor in backfill strength. The maximum range influential factor of the slump range is the slurry mass concentration, which is 6.36. The influential factor of the minimum range is the cement-sand ratio, which is 1.56, which shows that the mass concentration of the slurry affects the slump of the filling slurry. The effect of the cement-sand ratio has the least on slurry slump. The influential factors sequence in the slump of filling slurry are slurry mass concentration > waste rock content > cement-sand ratio, and slurry mass concentration is the main influential factor of the slump of filling slurry. Cement-sand ratio is the most important factor affecting backfill strength. The slurry mass concentration is the most important factor affecting backfill slurry fluidity, which is consistent with Wang’s findings [6].

### 3.2. Interactive Influence of Multi-Factors on Backfill Strength and Slump

RSM experiment results (Table 8) were obtained by multivariate nonlinear fitting, and response models of the compressive strength and slurry slump in slurry mass concentration, waste rock content and cement-sand ratio of ultra-fine tailings backfill at different curing ages were established, as shown in Equations (2)–(4). The correlation coefficient square *R*_1_^2^ of *y*_1_, *R*_2_^2^ of *y*_2_ and *R*_3_^2^ of *y*_3_ for nonlinear model value are 0.9940, 0.9945 and 0.9912, respectively. The results show that the three models have high reliability.
*y*_1_ = 27.3 − 0.835*x*_1_ + 0.0279*x*_2_ − 21.92*x*_3_ + 0.00621*x*_1_^2^ − 0.000671*x*_2_^2^ + 12.77*x*_3_^2^ + 0.000331*x*_1_*x*_2_ + 0.3315*x*_1_*x*_3_ + 0.0078*x*_2_*x*_3_
(*R*_1_^2^= 0.9940)(2)
*y*_2_ = 114−2.967*x*_1_ − 0.4951*x*_2_ − 79.1*x*_3_ + 0.0195*x*_1_^2^ + 0.000423*x*_2_^2^ + 31.78*x*_3_^2^ + 0.0063*x*_1_*x*_2_ + 0.9735*x*_1_*x*_3_ + 0.2241*x*_2_*x*_3_(*R*_2_^2^= 0.9945)(3)
*y*_3_ = −1108.8 + 31.01*x*_1_ + 4.325*x*_2_ + 306.4*x*_3_ − 0.2198*x*_1_^2^ − 0.02193*x*_2_^2^ − 77.9*x*_3_^2^ − 0.035*x*_1_*x*_2_ − 3.264*x*_1_*x*_3_ − 1.422*x*_2_*x*_3_
(*R*_3_^2^= 0.9912)(4)
where *y*_1_ is backfill uniaxial compressive strength for 7-day, MPa; *y*_2_ is 28-day backfill uniaxial compressive strength, MPa; *y*_3_ is slurry slump, cm; *x*_1_ is mass concentration of filling slurry, %; *x*_2_ is waste rock content, %; *x*_3_ is cement-sand ratio.

To verify that the effectiveness of multivariate nonlinear fitting models established based on response surface method, variance analysis is adopted on the established response model, as shown in Table 9.

Table 9 shows that the minimum value of *F* for the three models is *F* = 88.02 > *F*_0.05_(3,13) = 3.41, indicating that the three models are reliable and statistically reliable, and can effectively reflect the relationship between response values and influential factors. The significance tests of the three models were *p* < 0.001, indicating extremely significant and high reliability of the three models. Equations (2)–(4) show that ofr in the 7-day backfill strength *y*_1_ and 28-day backfill strength *y*_2_ models, the *p* value of waste rock content *x*_2_ is higher than the mass concentration *x*_1_ of backfill slurry and the cement-sand ratio *x*_3_. This implies that the waste rock content *x*_2_ has less influence on the 7-day strength *y*_1_ and 28-day strength *y*_2_ of the backfill compared with the filling slurry mass concentration *x*_1_ and the cement-sand ratio *x*_3_. In the 7-day backfill strength *y*_1_ model, the interaction term *x*_1_*x*_2_ (*p* = 0.734) between slurry mass concentration and waste rock content, the interaction term *x*_1_*x*_3_ (*p* = 0.003) and the interaction term *x*_2_*x*_3_ (*p* = 0.806) between slurry mass concentration and cement-sand ratio have the lowest *p* value. The results suggest that the interaction between the slurry mass concentration *x*_1_ and the cement-sand ratio *x*_3_ has the highest influence on the 7-day strength. In the backfill slump *y*_3_ model, the *p* values of each item in the model are ≤0.001, and only the cement-sand ratio *x*_3_ (*p* = 0.227) and *x*_3_^2^ (*p* = 0.018) of the model are greater than 0.001, indicating that the cement-sand ratio *x*_3_ has little influence on the slump.

The response surface diagram intuitively displays the influence of the interaction of different influential factors on response quantity and constructs the response surface diagram of response quantity and influential factors. The results are shown in Figure 5, Figure 6 and Figure 7.

Figure 5 and Figure 6 show that, under the action range of each level of waste rock content and slurry mass concentration, the fluctuation range of 7-day strength and 28-day strength is about 0~1 MPa, indicating that the interaction between waste rock content and slurry mass concentration has little influence on the 7-day and 28-day backfill strength. Under the interaction of slurry mass concentration and cement-sand ratio, the maximum backfill strength reaches 2.5 MPa at 7 days and 3.5 MPa at 28 days, respectively, indicating that the interaction of slurry mass concentration and cement-sand ratio has the highest influence on backfill strength. Under the interaction of the cement-sand ratio and waste rock content, it can be seen that the waste rock content level has little influence on the 7-day and 28-day backfill strength. As can be seen from Figure 7, under the interaction of slurry mass concentration and waste rock content, the variation range of slurry slump is 19~28 cm. Under the interaction of slurry mass concentration and cement-sand ratio, the variation range of slurry slump is 18~26 cm. Under the interaction of cement-sand ratio and waste rock content, the variation range of slurry slump is 22~28 cm. It can be seen that the interaction between slurry mass concentration and waste rock content and the interaction between slurry mass concentration and cement-sand ratio have the highest influence on the slump of the filling slurry, and the slurry mass concentration is the main influential factor in the slump of the filling slurry. This conclusion is consistent with the range analysis conclusion of the orthogonal test. At the same time, the interaction between slurry mass concentration and cement-sand ratio has the greatest influence on backfill strength, which is also supported by Huang’s similar work [16].

### 3.3. Multi-Response Robust Optimization of Filling Proportion

(1)Robust optimization of filling proportion

Based on the traditional desirability function method, the robust optimization desirability function method [26] reduces the sensitivity of the target to noise factors by selecting the level combination of controllable factors to improve the robustness of the target and make the target performance more stable and reliable. The method takes the range *R*_i_(*i* =1, 2 … *m*) to convert a specific robust desirability function *d*_ri_, with the value ranging from 0~1. The higher the value, the more satisfactory the response surface function is. The single desirability function of the traditional desirability function method is:(5)$$d_{\mathrm{si}}=\left\{\begin{array}{c} {0\;y}_{i}<L_{i}\\ {\left(\frac{y_{i}{-L}_{i}}{T_{i}{-L}_{i}}\right)}^{t}{\;L}_{i}\leq y_{i}\leq T_{i}\\ {1\;y}_{i}>T_{i} \end{array}\right.$$
(6)$$d_{\mathrm{si}}=\left\{\begin{array}{c} {1\;y}_{i}<T_{i}\\ {\left(\frac{U_{i}{-y}_{i}}{U_{i}{-T}_{i}}\right)}^{t}{\;T}_{i}\leq y_{i}\leq U_{i}\\ {0\;y}_{i}>U_{i} \end{array}\right.$$
(7)$$d_{\mathrm{si}}=\left\{\begin{array}{c} {0\;y}_{i}<L_{i}\\ {\left(\frac{y_{i}{-L}_{i}}{T_{i}{-L}_{i}}\right)}^{t_{1}}{\;L}_{i}\leq y_{i}\leq T_{i}\\ {\left(\frac{U_{i}{-y}_{i}}{U_{i}{-T}_{i}}\right)}^{t_{2}}{\;T}_{i}\leq y_{i}\leq U_{i}\\ {0\;y}_{i}>U_{i} \end{array}\right.$$
where *T*_i_, *U*_i_ and *L*_i_ are set target value, maximum value and minimum value of the optimization value interval for the response amount of *i*; *t* is a constant reflecting the importance of a single response relative to robustness. Equation (5) applies to the high desirability characteristic, i.e., a higher response variable value and a traditional desirability value imply a better optimization effect. Equation (6) is suitable for low desirability characteristics, i.e., a lower response variable value and higher traditional desirability value imply a better optimization effect. Equation (7) is applicable to target desirability characteristics; the closer the response value is to the set target value, the greater the traditional desirability value, indicating that the optimization effect is better.

The overall traditional desirability function is:(8)$$D_{\mathrm{optimization}}={\left(d_{s1}d_{s2}{\ldots \ldots d}_{\mathrm{sm}}\right)}^{1/m}$$
where *m* is the number of the response quantity.

In the robust desirability function method, a single robust desirability function is:(9)$$d_{\mathrm{ri}}=\left\{\begin{array}{c} {0\;R}_{i}\geq T_{i}{-L}_{i}\\ {(1-\frac{R_{i}}{T_{i}{-L}_{i}})}^{s}\;0<R_{i}<T_{i}{-L}_{i}\\ {1\;R}_{i}=0\; \end{array}\right.$$
(10)$$d_{\mathrm{ri}}=\left\{\begin{array}{c} {0\;R}_{i}\geq {U-T}_{i}\\ {(1-\frac{R_{i}}{U_{i}{-T}_{i}})}^{s}\;0<R_{i}<U_{i}{-T}_{i\;}\\ {1\;R}_{i}=0\; \end{array}\right.$$
(11)$$d_{\mathrm{ri}}=\left\{\begin{array}{c} {0\;R}_{i}\geq U_{i}{-L}_{i}\\ {(1-\frac{R_{i}}{U_{i}{-L}_{i}})}^{s}\;0<R_{i}<U_{i}{-L}_{i\;}\\ {1\;R}_{i}=0\; \end{array}\right.$$
where *s* is a constant reflecting the importance of a single response relative to robustness. Equation (9) applies to the high desirability characteristic, i.e., a higher response variable value and a smaller range of the response value implies a better robust desirability value. Equation (10) is suitable for low desirability characteristics, i.e., a lower response variable value and higher range of the response value implies a better robustness desirability value. Equation (11) is applicable to target desirability characteristics; the closer the response value is to the target value, the greater the robust desirability value is.

The overall robust desirability function is:(12)$$D_{\mathrm{robustness}}={\left(d_{r1}d_{r2}{\ldots \ldots d}_{\mathrm{rm}}\right)}^{1/m}$$

The robust optimization desirability function weighing optimality and robustness is:(13)$$D_{\mathrm{overall}}{=(D}_{\mathrm{optimization}}{)}^{w_{1}}{{(D}_{\mathrm{robustness}})}^{w_{2}}$$
where *D*_optimization_ indicates a traditional desirability function, *D*_robustness_ indicates a robust desirability function, *w*_1_ and *w*_2_ indicate weights ascribe to optimality and robustness respectively. The weights *w*_1_ and *w*_2_ can be selected according to the relative importance of optimality and robustness.

First, the robust desirability function method takes slurry mass concentration, waste rock content and cement-sand ratio as influential factors, and 7-day, 28-day backfill strength and slurry slump as a target. The target was obtained by RSM tests and a response surface regression model between influential factors. On the basis of the regression model, desirability function *d*_si_ to be built through Equations (5)–(7). The single desirability function is calculated under different influential factors and different target values are converted into numbers within the range of 0~1. Then, the optimal solution, Equation (8), is sought through the overall traditional desirability function.

Second, the single robust desirability function *d*_ri_ through Equations (9)–(11) is constructed. The single robust desirability function is calculated under different influential factors and different target values are converted into numbers within the range of 0~1. The robust overall desirability function is established by Equation (12).

Finally, the optimal solution is obtained by calculating the desirability function using Equation (13) of robust optimization, which balances the optimality and robustness through the optimization algorithm to achieve the purpose of multi-objective robust optimization. Multi-response robust parameter optimization needs to set the optimization interval of the response value.

The filling slurry needs to meet the requirements of gravity transportation in the Shuiyindong gold mine; that is, the slump range is controlled at 26~29 cm to get the appropriate fluidity standard of the filling slurry [27,28]. According to the “Code for design of nonferrous metal mining (GB50771-2012) and field requirements for the first-step of open stoping with subsequent filling method, the backfill strength is required to be no less than 1.0 MPa for 7-day and no less than 2.0 MPa for 28 days. Slump is the target characteristic (response optimization target value). Under the condition of meeting the requirements of filling slurry transportation, the optimal target value of the slump is 27.5 cm. The 7-day strength range of backfill is set at the test range of 0.244 MPa~2.585 MPa, and the 28-day strength range is set at the test range of 0.238 MPa~3.695 MPa. Considering that the strength is optimal under the condition of filling self-flow transportation, backfill strength is a high desirability characteristic. Considering the stability and optimality of optimization parameters comprehensively, the weights of *w*_1_ and *w*_2_ are both 0.5; that is, the optimality and robustness are of the same importance.

According to Equations (2)–(4), the single desirability function value under different influential factors is obtained, among which the single desirability function value of the 7-day strength was 0.7450. The single desirability function of the 28-day strength was 0.6809. The single desirability function value of the slump is 0.9759, and the overall traditional desirability function value calculated by Equation (5) is 0.7911. Then, a single robust desirability function was constructed according to Equations (6)–(8). The single robust desirability function value of the 7-day strength was 0.8133. The single robust desirability function of the 28-day strength was 0.7988. The single robust desirability function value of the slump is 0.9212, and the overall robust desirability function value calculated by Equation (9) is 0.8427. Finally, the robust optimization desirability function value that balances optimality and robustness is 0.8165, calculated by Equation (10), which is close to 1, indicating that the optimization effect is reliable. The desirability function value obtained by the traditional method is generally 0.4~0.5 [29]. The result obtained by the robust desirability function method is greater than this value, indicating that the optimization effect is reliable.

According to the robust optimization desirability function method, when the slurry mass concentration is 68.36%, the waste rock content is 36.72%, and the cement-sand ratio is 1:3, the backfill strength reaches the optimal level at the same time on the 7-day and 28-day of multiple response targets. The slump of the filling slurry is close to the target value of 27.5 cm, and the parameters of the filling material ratio are robust and optimal. The overall robust optimal desirability was 0.8165. The optimized backfill strength was 1.988 MPa for 7 days, 2.592 MPa for 28 days, and 27.3 cm for slump.
(2)Multi-response robust optimization validation test

To verify that the backfill strength obtained by the robust optimization desirability function method meets the requirements, a laboratory test was carried out with slurry mass concentration of 68.36%, waste rock content of 36.72% and cement-sand ratio of 1:3. The test results are shown in Table 10. The porosity of this proportional mixture was 21.3%. Porosity is a result of the particle distribution and packing density of backfill slurry, and it can also be indicated by slurry mass concentration. A higher packing density of slurry is usually obtained with tight aggregates of fine and coarse particles and therefore has a lower porosity. However, this well-aggregated slurry tends to accelerate the hydration reaction and cementation of backfill materials, which is beneficial for improving backfill strength. The moderate packing density of the optimal standard backfill sample (2.23 g/cm^3^) achieves equilibrium in backfill strength and slurry fluidity and meets the onsite filling requirements.

Verification test results show that the 7-day strength, 28-day strength and slump test values of the backfill are all within the error range of the calculated values. The 7-day backfill strength was 1.94 MPa ≥ 1.0 MPa, and the 28-day backfill strength was 2.56 MPa ≥ 2.0 MPa. The slump of 26.9 cm is within the self-flow slump range of 26 cm~29 cm. The verification test shows that the multi-response robust parameter optimization has high reliability, which achieves the self-flow transportation of slurry and the strength optimization of backfill.

## 4. Conclusions

To alleviate the low backfill strength and poor slurry fluidity of Carlin-type gold deposits with ultra-fine tailings, orthogonal experiments and the RSM method were used to reveal the influence of variable factors and their interaction on backfill strength and slurry slump. The influential factor sequence of backfill unconfined compressive strength at each curing age was the cement-sand ratio > slurry mass concentration > waste rock content. Backfill slurry slump was sequentially influenced by slurry mass concentration > waste rock content > cement-sand ratio.The interactive influence of multi-factors on the backfill strength and slump was studied by constructing a response surface with slurry mass concentration, waste rock content and cement-sand ratio. The results showed that the interaction between slurry mass concentration and cement-sand ratio has a positive correlation with backfill strength. The interaction between slurry mass concentration and cement-sand ratio, as well as the interaction between slurry mass concentration and waste rock content, contributed the highest effect on the slurry slump. Slurry mass concentration is a dominant influential factor in slurry slump, which verifies the reliability of range analysis derived from the orthogonal experiment.The filling proportion with satisfactory slurry fluidity is optimized by multi-response robust optimization for Carlin-type gold deposits with ultra-fine tailings. The validation test shows that the cemented filling material proportion is optimal with 68.36% slurry mass concentration, 36.72% waste rock content and 1:3 cement-sand ratio. With this filling proportion, the 7-day and 28-day backfill strengths were 1.94 MPa and 2.56 MPa, respectively, and the slurry slump was 27.5 cm.

## Figures and Tables

**Figure 1 materials-15-06902-f001:**
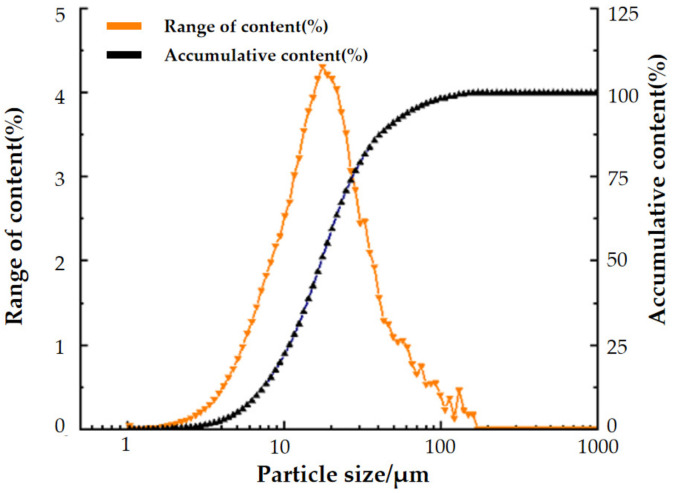
Distribution of ultra-fine tailings in the Shuiyindong gold mine.

**Figure 2 materials-15-06902-f002:**
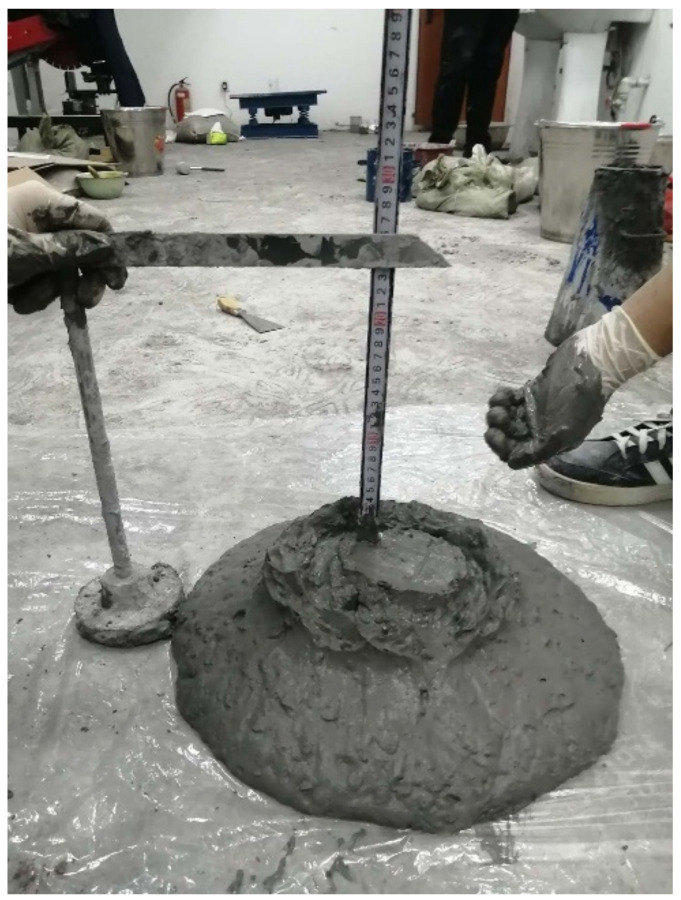
Slurry slump measuring using slump cylinder and tape measure with 72% slurry mass concentration, 35% waste rock content and 1:4 cement-sand ratio.

**Figure 3 materials-15-06902-f003:**
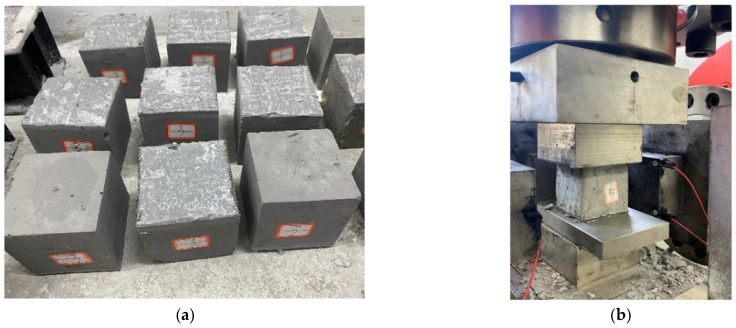
Backfill (**a**) specimen preparation and (**b**) unconfined compressive strength test of backfill.

**Figure 4 materials-15-06902-f004:**
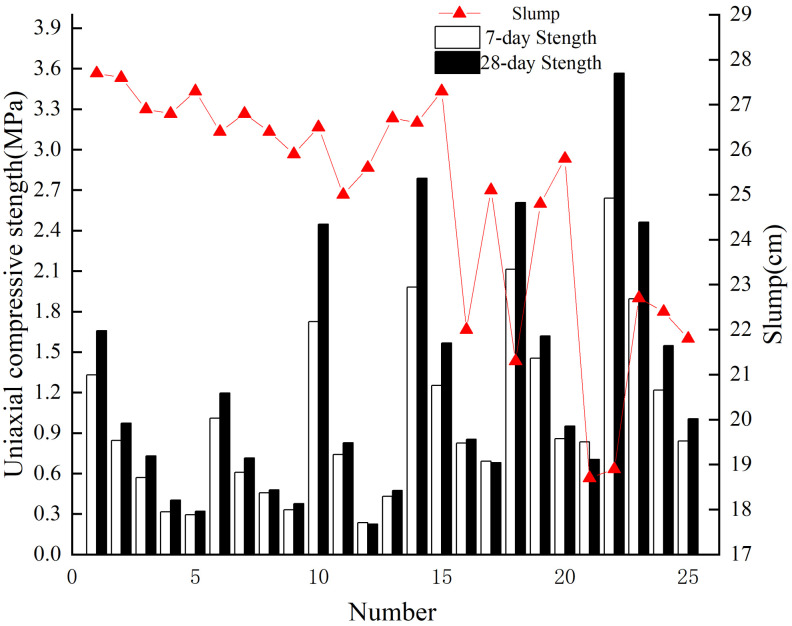
Influence of single factor on strength and slump of backfill by orthogonal ratio test.

**Figure 5 materials-15-06902-f005:**
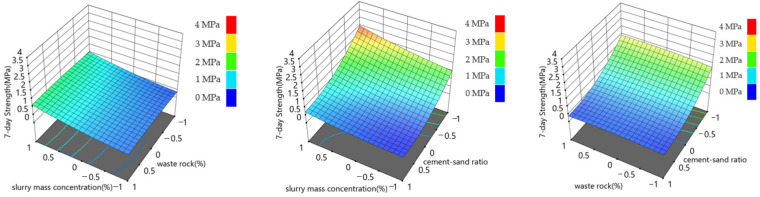
Relationship between backfill strength and backfill mixture component ratio after 7 days.

**Figure 6 materials-15-06902-f006:**
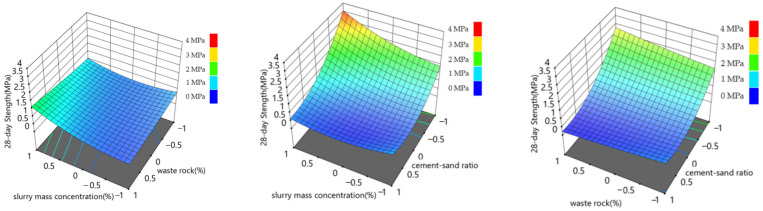
Relationship between backfill strength and backfill mixture component ratio after 28 days.

**Figure 7 materials-15-06902-f007:**
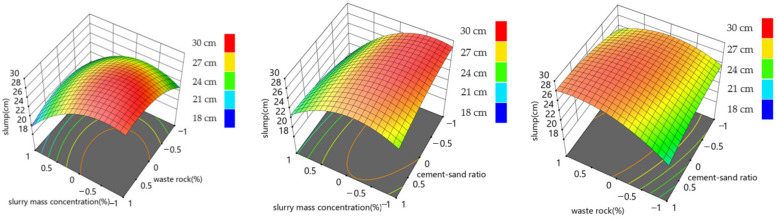
Relationship between the slump and the ratio of the components of the backfilling mixture.

**Table 1 materials-15-06902-t001:** Waste rock particle size composition.

Particle Size/mm	wt/%	Particle Size/mm	wt/%	Particle Size/mm	wt/%
−5	2.23	20–40	12.24	150–200	21.96
5–10	3.06	40–80	14.53	200–300	25.34
10–20	7.44	80–150	2.15	+300	11.03

**Table 2 materials-15-06902-t002:** Chemical composition of ultra-fine tailings (mass fraction)/%.

Component	Au	Fe	Cao	MgO	Al_2_O_3_	SiO_2_	S	K	Mn	Si	Loss on Ignition
Content/%	0.00012	5.54	12.37	6.88	14.45	37.22	2.98	4.99	5.23	9.34	0.99988

**Table 3 materials-15-06902-t003:** Level of orthogonal test of filling proportion influenced by a single factor on strength and slump of backfill.

Slurry Mass Concentration A (%)	Waste Rock Content B (%)	Cement-Sand Ratio C
64	25	1:3
66	30	1:4
68	35	1:6
70	40	1:8
72	45	1:10

**Table 4 materials-15-06902-t004:** Orthogonal test scheme of filling proportion influenced by a single factor on strength and slump of backfill.

Run	Slurry Mass Concentration (%)	Waste Rock Content (%)	Cement-Sand Ratio	Run	Slurry Mass Concentration (%)	Waste Rock Content (%)	Cement-Sand Ratio
1	64	25	1:3	14	68	40	1:3
2	64	30	1:4	15	68	45	1:4
3	64	35	1:6	16	70	25	1:8
4	64	40	1:8	17	70	30	1:10
5	64	45	1:10	18	70	35	1:3
6	66	25	1:4	19	70	40	1:4
7	66	30	1:6	20	70	45	1:6
8	66	35	1:8	21	72	25	1:10
9	66	40	1:10	22	72	30	1:3
10	66	45	1:3	23	72	35	1:4
11	68	25	1:6	24	72	40	1:6
12	68	30	1:8	25	72	45	1:8

**Table 5 materials-15-06902-t005:** The interaction of multi-factors on backfill strength and slump influential factors and level of RSM test of backfill proportion.

Influential Factor	Coding Level		
	−1	0	1
Slurry mass concentration (%)	64	68	72
Waste rock content (%)	25	35	45
Cement-sand ratio	1:3	1:6	1:10

**Table 6 materials-15-06902-t006:** RSM experiment scheme of filling proportion is influenced by multi-factor interaction on strength and slump of backfill.

Run	Coded Variables			Original Variables		
	Slurry Mass Concentration (%)	Waste Rock Content (%)	Cement-Sand Ratio	Slurry Mass Concentration (%)	Waste Rock Content (%)	Cement-Sand Ratio
1	0	0	0	68	35	1:6
2	1	−1	0	72	25	1:6
3	0	1	−1	68	45	1:3
4	0	0	0	68	35	1:6
5	1	1	0	72	45	1:6
6	0	1	1	68	45	1:10
7	1	0	1	72	35	1:10
8	−1	1	0	64	45	1:6
9	−1	−1	0	64	25	1:6
10	0	0	0	68	35	1:6
11	0	−1	1	68	25	1:10
12	0	0	0	68	35	1:6
13	1	0	−1	72	35	1:3
14	−1	0	1	64	35	1:10
15	0	−1	−1	68	25	1:3
16	0	0	0	68	35	1:6
17	−1	0	−1	64	25	1:3

**Table 7 materials-15-06902-t007:** Range analysis of influence of single factor on strength and slump of backfill in orthogonal ratio test.

Level of Factors	The Influence of Single Factor on 7-Day Strength Range			The Influence of Single Factor on 28-Day Strength Range			The Influence of Single Factor on Slump Range		
	Slurry Mass Concentration Means (%)	Waste Rock Content Means (%)	Cement-Sand Ratio	Slurry Mass Concentration Means (%)	Waste Rock Content Means (%)	Cement-Sand Ratio	Slurry Mass Concentration Means (%)	Waste Rock Content Means (%)	Cement-Sand Ratio
1	0.671	0.948	1.958	0.816	1.048	2.613	27.26	23.96	24.20
2	0.826	1.004	1.291	1.043	1.231	1.564	26.40	24.80	25.76
3	0.928	1.093	0.799	1.176	1.351	0.954	26.24	24.80	25.38
4	1.188	1.061	0.535	1.343	1.347	0.593	23.80	25.30	24.52
5	1.485	0.994	0.516	1.857	1.258	0.511	20.90	25.74	24.74
Range	0.814	0.145	1.442	1.041	0.303	2.102	6.36	1.78	1.56

**Table 8 materials-15-06902-t008:** Results of multi-factor interaction on backfill strength and slump in the RSM test.

Number	Slurry Mass Concentration (%)	Waste Rock Content (%)	Cement-Sand Ratio	7-Day Strength (MPa)	28-Day Strength (MPa)	Slump (cm)
1	0	0	0	0.732	0.633	28.5
2	1	−1	0	0.904	0.919	19.2
3	0	1	−1	1.897	2.829	24.7
4	0	0	0	0.744	0.655	28.4
5	1	1	0	1.119	1.515	20.6
6	0	1	1	0.337	0.238	27.5
7	1	0	1	0.813	0.792	22.6
8	−1	1	0	0.567	0.550	29.1
9	−1	−1	0	0.405	0.962	22.1
10	0	0	0	0.703	0.628	28.4
11	0	−1	1	0.387	0.656	23.2
12	0	0	0	0.714	0.632	28.2
13	1	0	−1	2.585	3.695	19.6
14	−1	0	1	0.244	0.641	25.4
15	0	−1	−1	1.830	2.127	26.4
16	0	0	0	0.689	0.613	28.8
17	−1	0	−1	1.475	1.801	28.9

**Table 9 materials-15-06902-t009:** Analysis of variance of response surface regression model for influence of multi-factor interaction on backfill strength and slump.

Source of Variation	Sum of Squares			Mean Square			*F* Value			*p* Value		
	*y* _1_	*y* _2_	*y* _3_	*y* _1_	*y* _2_	*y* _3_	*y* _1_	*y* _2_	*y* _3_	*y* _1_	*y* _2_	*y* _3_
Model	6.540	14.021	203.702	0.727	1.558	22.634	129.812	324.084	88.022	0.000	0.000	0.000
*x* _1_	1.034	1.617	87.745	1.034	1.617	87.745	184.773	336.457	341.233	0.000	0.000	0.000
*x* _2_	0.019	0.097	7.677	0.019	0.097	7.677	3.465	20.123	29.864	0.105	0.003	0.001
*x* _3_	4.509	8.252	0.451	4.509	8.252	0.451	805.483	1716.683	1.751	0.000	0.000	0.227
*x* _1_ ^2^	0.042	0.410	48.459	0.042	0.410	48.459	7.447	85.282	188.453	0.029	0.000	0.000
*x* _2_ ^2^	0.019	0.008	22.614	0.019	0.008	22.614	3.383	1.575	87.946	0.108	0.251	0.000
*x* _3_ ^2^	0.077	0.479	2.422	0.077	0.479	2.422	13.825	99.628	9.426	0.007	0.000	0.018
*x* _1_ *x* _2_	0.001	0.254	7.840	0.001	0.254	7.840	0.132	52.849	30.494	0.734	0.000	0.001
*x* _1_ *x* _3_	0.105	0.901	14.162	0.105	0.901	14.162	18.663	187.425	55.073	0.003	0.000	0.000
*x* _2_ *x* _3_	0.000	0.298	12.010	0.000	0.298	12.010	0.073	62.073	46.706	0.806	0.000	0.000

**Table 10 materials-15-06902-t010:** Results of laboratory tests of robust optimization parameters.

7-Day Strength (MPa)		28-Day Strength (MPa)		Slump (cm)	
calculated value	experimental value	calculated value	experimental value	calculated value	experimental value
1.988 ± 0.05	1.940	2.592 ± 0.05	2.555	27.3 ± 0.5	26.9

## Data Availability

The data presented in this study are available on request from the corresponding author.

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
