# Peer review of "Multi-Response Robust Parameter Optimization of Cemented Backfill Proportion with Ultra-Fine Tailings"

_materials, 2022, doi:10.3390/ma15196902_

Round 1

Reviewer 1 Report

The presented article is devoted to solving the important problem of selecting the optimal composition of the backfilling mixture in conditions when ultra-fine tailings are formed in the mine during enrichment, which serve as the main component. This component reduces the strength of the backfill and makes it difficult to transport the mixture into the goaf. There is an urgent need to study the effect of backfilling mixture components on the strength and slump. The authors applied a combined methodological approach consisting of factor analysis, robust optimization method and laboratory studies. It is impressive that the authors compared the analytical and experimental results of the study of the strength and slump of the backfill with a change in the ratio of components, which showed high reliability of the analytical model.

The article undoubtedly has scientific and practical value. The subject of the article corresponds to the journal. I have a good impression of the quality of the research.

However, after a detailed acquaintance with the research material, I had several comments and recommendations to improve the quality of the article.

More details:

1. In my opinion, the introduction is too extensive. It is recommended to revise the structure of the introduction. In the introduction, the problems of the research topic should be disclosed. The text in lines 68-89, which provides a description of the RSM method, should be moved to the beginning of subsection 2.3.

2. The introduction also vaguely expressed the unexplored area. Authors, please add specifically what was not studied before you or insufficiently studied after the analysis (line 67).

3. In subsection 2.1 please explain what is the waste rock size range in the dump and why did you choose -5 mm? How will the fraction be separated from the mixture of rocks? Will there be additional costs? Also indicate the type of rock in the dump.

4. In Figure 1, decipher the graphs of different colors.

5. Please provide the appropriate citation after the text in lines 131-133.

6. Please indicate at the end of subsection 2.4 how many experimental backfill samples were prepared. Also, I would recommend in subsection 2.4 to provide several photographs (if any) from the process of preparing backfilling mixtures and testing them (mixing, measuring slump and strength). This will increase the understanding of conducting experimental studies.

7. Please place tables 6 and 7 inseparably. The title of subsection 2.3 should be moved to a new page. Move the caption to figure 1 to the page above.

8. I would recommend calling Figures 3 and 4 “Relationship between backfill strength and backfill mixture component ratio after 7(28) days“. Figure 5 – “Relationship between the slump and the ratio of the components of the backfilling mixture“. (or similar names as edited by the authors). The proposed name will more accurately convey the content of the graphs.

9. References in the introduction are made only to Chinese researchers. Dear authors, please take into account the work of other scientists and expand the geography of citation in the introduction (5-10 references). Almost all sources of literature are Chinese. Your article has been submitted to the reputable international journal Materials. This will show respect for researchers from other countries.

The following scientific articles can be considered by the authors:

Petlovanyi, M., & Mamaikin, O. (2019). Assessment of an expediency of binder material mechanical activation in cemented rockfill. ARPN Journal of Engineering and Applied Sciences, 14(20), 3492-3503.

Mashifana, T., & Sithole, T. (2021). Clean production of sustainable backfill material from waste gold tailings and slag. Journal of Cleaner Production, 308, 127357. https://doi.org/10.1016/j.jclepro.2021.127357

Petlovanyi, M.V., Zubko, S.A., Popovych V.V., & Sai, K.S. (2020). Physicochemical mechanism of structure formation and strengthening in the backfill massif when filling underground cavities. Voprosy Khimii i Khimicheskoi Tekhnologii, (6), 142-150. https://doi.org/10.32434/0321-4095-2020-133-6-142-150

Reviewer 2 Report

Dear author,

Line 37, review spaces.

Table 1. Does table 1 include the complete composition?. If not, fill it out.

Do the measurements presented in tables have a standard deviation?. Please, include it.

Table 8. The results should be presented with same significant figures.

Please, review the experimental methodology.

Regards,

Reviewer 3 Report

document present an original approch to estimate the influence of mixture design on the rheological and strenght of slurry Globaly the document is original and structured. I suggeget to accept for publication in materials journal after minor correction. 

1. In fig.1, the max value of total content should be correct. Total content of particle can't exceed 100%

2. Please provide more information about the composition of used cement 

3. Strenght and flow depend mainly on total porosity and packing density of the mix. Can you including this on discussion 

4. Are there any specifications or standards for the use of these materials min/max UCS and workability requirements. Its very important to present this on methods part

5. You should compare you results with previous results of litteratures. 
